# Deep Multilabel Multilingual Document Learning for Cross-Lingual Document Retrieval

**DOI:** 10.3390/e24070943

**Published:** 2022-07-07

**Authors:** Kai Feng, Lan Huang, Hao Xu, Kangping Wang, Wei Wei, Rui Zhang

**Affiliations:** 1College of Computer Science and Technology, Jilin University, Changchun 130012, China; fengkai17@mails.jlu.edu.cn (K.F.); huanglan@jlu.edu.cn (L.H.); xuhao@jlu.edu.cn (H.X.); wangkp@jlu.edu.cn (K.W.); 2School of International Economics and Trade, Changchun University of Finance and Economics, Changchun 130012, China; weiweiccr@126.com

**Keywords:** cross-lingual document retrieval, cross-lingual features, cross-lingual document representation

## Abstract

Cross-lingual document retrieval, which aims to take a query in one language to retrieve relevant documents in another, has attracted strong research interest in the last decades. Most studies on this task start with cross-lingual comparisons at the word level and then represent documents via word embeddings, which leads to insufficient structure information. In this work, the cross-lingual comparison at the document level is achieved through the cross-lingual semantic space. Our method, MDL (deep multilabel multilingual document learning), leverages a six-layer fully connected network to project cross-lingual documents into a shared semantic space. The semantic distances can be calculated when the cross-lingual documents are transformed into embeddings in semantic space. The supervision signals are automatically extracted from the data and then used to construct the semantic space via a linear classifier. The ambiguity of manual labels could be avoided and the multilabel supervision signals can be acquired instead of a single label. The representation of the semantic space is enriched by multilabel supervision signals, which improves the discriminative ability of the embeddings. The MDL is easy to extend to other fields since it does not depend on specific data. Furthermore, MDL is more efficient than the models training all languages jointly, since each language is trained individually. Experiments on Wikipedia data showed that the proposed method outperforms the state-of-the-art cross-lingual document retrieval methods.

## 1. Introduction

With the rapid growth of multilingual information on the Internet, cross-lingual document retrieval is becoming increasingly important for search engines. Monolingual information retrieval will miss information in other languages. This could be very important, for example, users may want to find news in foreign languages for the same event. However, current search engines usually return documents written in the same language, discarding many valuable results written in other languages. The information retrieval task is a difficult problem because queries and documents are likely to use different vocabularies when looking for correlations between them. This is more obvious in the task of cross-lingual document retrieval, thus, how to represent and compare documents across language barriers has attracted a lot of research and attempts.

To tackle the issue of the language barrier, many translation-based methods have achieved good results in cross-lingual retrieval tasks in the past decades [1]. These methods translate queries or documents first and then use the monolingual retrieval method to rank the candidate documents. The retrieval performance is tied down by the machine translation method and lack of flexibility. On the one hand, as machine translation improves performance with high-resource corpora, the performance of cross-lingual document retrieval improves. On the other hand, the result of the retrieval task is particularly dependent on the translation quality, any translation errors and ambiguity from the source language or the target language will cause disasters for the retrieval results. Moreover, the amount of translation is always huge, and the cost of time and storage is always expensive [1]. Therefore, large-scale translation in the Internet environment is impractical, also for some low-resource languages or domains which they do not contain enough data for training the machine translator, a more lightweight document representation is urgently needed [2].

While, for the purpose of obtaining a more general cross-lingual document representation, many strategies have been proposed such as knowledge-base based approaches [3,4]. Using concept collections from a knowledge base to represent documents avoids a lot of computational overhead, while it would lose most structural information of the documents themselves. This type of approach is limited by the conceptual scope of the knowledge base. Especially when low-resource languages are included, the number of the concept intersections covering all languages is much smaller. It is a heuristic method, which does not fully consider the document structure and cannot accurately cover the meaning of the document [2,4]. Moreover, it is difficult to deal with words out of vocabulary, and at the same time, the document representation is not optimized via learning. There are also studies that combine speech features to improve the quality of multilingual document representations [5,6] and representing documents based on features of machine translation and automatic speech recognition (ASR). Speech features can enrich the semantics of documents, and thus enhance the expressiveness of document representation. However, these studies rely on speech corpora and the quality of speech recognition features.

Although most cross-lingual document representation methods rely on high-resources language data or parallel corpus, some studies have proved that it is effective to solve the cross-lingual document retrieval problem based on the comparable corpus [2,7,8]. It greatly alleviates the problem of resource scarcity. Most of these approaches achieve the cross-lingual at the lexical level first and then get the document embeddings, which is still a heuristic process.

Based on this observation, we propose a deep multilabel multilingual document learning method (MDL), addressing the problem of cross-lingual document learning as a multilabel classification problem by getting embeddings at the document level directly through the cross-lingual signals in the classification process. Multilingual documents are mapped to a shared semantic space as language-independent features, and the relevant scores are then calculated for the retrieval process. MDL performs cross-lingual optimization at the document level, rather than implementing cross-lingual vocabulary first and then obtaining document representations. The model first constructs a shared semantic space based on the multilabels from the data without adopting any additional cross-lingual information. The multilabels are automatically generated based on the latent Dirichlet assignment (LDA) [9] algorithm. We employ the unsupervised document embedding method doc2vec [10], which can contain the document structure information to obtain the initial document representations. The multilabel supervision signals are then used to train the language-specific encoders that contain the desired mapping relations between the document representation and the semantic space. In the testing stage, the cross-lingual documents are mapped into the semantic space by the encoders. Thus, the semantic distances of the cross-lingual documents can be calculated based on the semantic space. There are several benefits to doing this, first, it could contain language-unique structure information in the document representation process. Second, it could greatly reduce the amount of model computation, because the input during the training stage is no longer a collection of words but a collection of documents. The third is that the demand for a corpus is greatly reduced. A comparable corpus with document topic alignment is required, while the lexical aligned dictionaries and sentence aligned parallel corpora are no longer required. Contrary to other methods that involved all languages trained together, another advantage of MDL is that each language is trained separately. Therefore, MDL is easily extended to other languages without retraining existing languages. The main contributions of this work can be summarized as follows.

A framework for cross-lingual document embeddings through the multilabel classification process is proposed.A novel deep multilabel multilingual document learning architecture is proposed to reduce the difference between the distribution of documents in different languages. Since each language is trained separately and takes document vectors as input, the model is more efficient than the jointly trained models at the training stage.A cross-lingual retrieval framework based on document representation. We train the model on Wikipedia data in four languages with 800 k entries, and results demonstrate that it outperforms state-of-the-art methods on document retrieval tasks by more than 30%.

## 2. Related Work

With the popularity of pre-training methods and word embedding methods in the natural language processing (NLP) field, many cross-lingual word embeddings (CLWE) methods have also been proposed that have achieved a competitive cross-lingual retrieval performance in recent years [8,11,12]. Generally, cross-lingual word embedding methods require different supervision signals, including vocabulary alignment, sentence alignment, and document alignment [11,13]. Additionally, there are many unsupervised cross-lingual word embedding methods being studied [14,15,16]. These methods obtain the cross-lingual vocabulary through supervised signal or unsupervised strategy first, then represent documents through similar ways of word embeddings combination [13,16]. The structure of information in texts is not considered well and the embeddings are not optimized explicitly for the document level [2]. To improve the quality of cross-lingual word embeddings and reduce the level of supervision, many follow-up studies have focused on the representation of similarity between languages [13,17].

The spatial projection method was proposed to optimize cross-lingual word embeddings, which is a weakly supervised method [18]. It has been verified that this simple linear mapping can achieve good results, and there are many studies to follow this strategy [14]. The supervised method directly uses the existing dictionary, while the unsupervised method automatically builds the seed dictionary. Using a small number of initial dictionaries to get the vector space in which the two words are aligned, afterward, learn the projection of the conversion between the two spaces. This approach focuses on exploiting the similarity between word embedding spaces to learn this relationship [19].

Vulić and Moens obtained pseudo-bilingual documents by merging document-aligned corpora and obtained cross-lingual word embeddings based on the skip-gram model [8]. The work of Alexis et al. presents an unsupervised approach that achieves competitive results on word and sentence level retrieval problems, and this method also performs well on cross-lingual document retrieval tasks [14,19]. In short most of the current methods still rely on parallel corpora, in addition, it is still necessary to define document representation based on word embedding [20].

Most cross-lingual document embedding methods use alignment relationships to induce shared semantic spaces, which rely on a high-quality parallel corpus. In general scenarios, comparable corpora with topic alignment are more readily available than parallel corpora. Thus, approaches that require document-aligned, comparable data, prove promising as it significantly alleviates the resource scarcity problem.

One line of thought focuses on cross-lingual topic models, and most of them are based on the latent Dirichlet allocation (LDA) algorithm [9,21]. Some approaches use the word-aligned corpus where the topic model is achieved by optimizing the semantic distribution of words [22,23]. The disadvantage is that it is limited by multilingual vocabulary alignment resources [24]. Other studies are focusing on the document alignment corpus, which utilize large aligned corpora effectively and map multilingual documents to corresponding topic distributions through training [25,26,27,28]. The focus of these methods is on how to describe the same concept in multiple languages, while our approach is concerned more with establishing connections between multilingual documents and concepts.

Instead of using combined word embeddings to obtain documents, cross-lingual representation methods at the document level are also proposed and studied. Josifoski’s work [2] proposes to obtain the document representations by minimizing the gap between monolingual words and cross-lingual terminology. The topic tags are directly used as supervised signals to induce cross-lingual document embeddings. It is a sufficiently complex problem because the number of tags is millions. Cr5 (cross-lingual reduced-rank ridge regression), a framework based on a linear algorithm is proposed to split the classification weights matrix, which is highly efficient for the massive tags. Experiments show that this linear model achieves better performance than the baseline in document retrieval tasks. Consequently, we will use Cr5 as our main baseline. The Cr5 model could be seen as an enhanced cross-lingual word representation since the word could be a document is this stage. However, due to the use of the bag-of-words model, although the frequency of word occurrence is considered, the semantic position of the word is ignored, and it is difficult to consider well of the text structure. We propose a method for cross-lingual embeddings, which structures the problem in a multilabel classification setting and uses comparable corpus in an efficient and scalable manner.

## 3. Proposed Method

### 3.1. Cross-Lingual Document Embedding

We propose to address the problem of cross-lingual document embedding as a classification problem that focuses on the use of class labels and comparable data. Our goal is to find the mappings that map different document distributions to the same distribution. In other words, map language-specific document distributions into a shared semantic space. In this framework, a classifier-based method for finding the mappings is applied. We first introduce the definition of cross-lingual document learning and then show how to obtain language-independent document features in a multilabel classification manner in the following subsections.

Suppose *L* represent the collection of different languages, the *j*-th document of a language $l_{i}$ is represented as $x_{j}^{\left(l_{i}\right)},l_{i}\in L$, and the set of all the $n_{i}$ documents is represented as $X^{\left(l_{i}\right)}=\left\{x_{1}^{\left(l_{i}\right)},x_{2}^{\left(l_{i}\right)},x_{j}^{\left(l_{i}\right)},\ldots ,x_{n_{i}}^{\left(l_{i}\right)}\right\}$. Along with the class labels set $Y^{\left(l_{i}\right)}=\left\{y_{1}^{\left(l_{i}\right)},y_{2}^{\left(l_{i}\right)},y_{j}^{\left(l_{i}\right)},\ldots ,y_{n_{i}}^{\left(l_{i}\right)}\right\}$, and $y_{j}^{\left(l_{i}\right)}=\left[k_{1j}^{\left(l_{i}\right)},k_{2j}^{\left(l_{i}\right)},\ldots ,k_{Cj}^{\left(l_{i}\right)}\right],k\in \left\{0,1\right\}$ is a label vector, where *C* is the number of classes. Thus, for each class $k_{c},c\in C$, $k_{cj}^{\left(l_{i}\right)}=1$ if the document $x_{j}^{\left(l_{i}\right)}$ belongs to the *c*-th class, while $k_{cj}^{\left(l_{i}\right)}=0$ if not.

Relevance scores cannot be calculated directly for $x_{j}^{\left(l_{i}\right)}$ from different language $l_{i}$, because they come from different spaces and have different distributions. The main goal is to map $X^{\left(l_{i}\right)},l_{i}\in L$ into the same space so that they can be compared with each other. Thus, multilingual document learning is defined as finding the mappings for each language that maps $X^{\left(l_{i}\right)}$ to shared semantic space. The transformation function that provides the mapping relation is expressed as $f_{l_{i}}\left(x^{\left(l_{i}\right)},\Theta^{i}\right)\in R^{d}$, *d* is the shared space dimension, $\Theta^{i}$ indicates the parameters that need to be learned. For simplicity and clarity of discussion, we refer $f_{l_{i}}\left(x^{\left(l_{i}\right)},\Theta^{i}\right)$ as the encoder in the following. Different sources of *X* require specific encoders for the same *Y*. Figure 1 shows the distributions of documents in four languages, including English, Italian, Danish, and Vietnamese. Each language has a different distribution. The shared semantic space is constructed by the same supervision signals. Additionally, each document distribution $X^{\left(l_{i}\right)},l_{i}\in L$ can be mapped into the shared semantic space through a language-specific encoder. The semantic distances can be calculated when multilingual documents are mapped as embeddings in the same semantic space. Thus, the goal of cross-lingual document embedding is to find the appropriate encoder for each language.

### 3.2. Shared Semantic Space Constructing

We construct the shared semantic space through the training process of the classification problem. The goal is to find the mappings which can be provided by a linear classifier. Generally, for a vector of inputs $x\in R^{p}$ and a predicted vector of labels $y\in R^{C}$, the classification process is to find the transformation relationship *W* and *b*, based on the “winner-takes-all” decision rule to make the label prediction more accurate as Equation (Equation 1) shows.
(1)$$y\left(x\right)=\underset{k\in \{1\ldots K\}}{\arg\max}W_{k}^{T}x+b_{k}$$

The semantic space comes from the decomposition of the transformation matrix $W\in R^{C\times p}$. It is easy to observe that the matrix *W* can be decomposed into the product of two matrices W=HΦ, $H\in R^{C\times r}$ and $\Phi \in R^{r\times p}$. Bring them into Equation (Equation 1) to get the following,
(2)$$y\left(x\right)=\underset{k\in \{1\ldots K\}}{\arg\max}H\left(\Phi x\right)+b_{k}$$
which can be regarded as transforming *x* into *r*-dimensional space through matrix Φ first, and then completing the classification task through matrix *H* which represents the linear relationship between data features and labels. Assuming that the data features $x^{\prime }$ and label *y* is given, then through such supervised training, *H* can be gradually optimized to improve the prediction accuracy. Similarly, assuming that the matrix *H* and the label *y* are fixed, by improving the data feature $x^{\prime }$, the prediction accuracy can also be improved in an iterative manner. In other words, the *r*-dimensional space is supervised by the category labels when *H* is fixed, and this space is the so-called semantic embedding space. The reason why the *r*-dimensional space can be used as the embedding space of the document is that this linear classification rule will guide the points of the same category to be close to each other in the embedding space, and the points of different categories are far away from each other. At the same time, in order to make the shared space more discriminative, *H* can be constrained to be an orthogonal matrix, which will guide the orthogonality between different categories in the shared space and make the data more discriminative.

The matrix Φ maps *x* to *r*-dimensional space and converts it to $x^{\prime }$. When the matrix Φ is regarded as an encoder, it means that the encoder can map *x* to the semantic space to get $x^{\prime }$. To sum up, suppose an encoder $f^{\left(l_{i}\right)}\left(x,\Theta^{i}\right)$, an orthogonal matrix *H*, $\Theta^{i}$ is a learnable parameter, then the prediction of the label Equation (Equation 2) becomes
(3)$$y\left(x\right)=Hf^{\left(l_{i}\right)}\left(x,\Theta^{i}\right),$$
and objective function as follows.
(4)$$J^{\left(l_{i}\right)}={∥Hf^{\left(l_{i}\right)}\left(x,\Theta^{i}\right)-y∥}_{2}$$

The encoder projects the document into an *r*-dimensional embedding representation. Equation (Equation 3) shows that the predicted label can be obtained by multiplying the *r*-dimensional vector by *H*. In other words, this *r*-dimensional space is linearly related to the label space. If the same labels are used as supervision signals, then texts in different languages can be mapped into the same space. Thereby, the correlation of multilingual texts can be calculated for the retrieval task.

### 3.3. Deep Multilabel Multilingual Document Learning

Note that the projection function is influenced by the input data and supervisory signals, especially the class labels are critical to the projection quality. It is practical to transform the projection problem into a single-label classification problem, where document-level mapping is achieved through a many-to-one category relationship. However, the category labels of such methods are usually one-hot representations with only one dimension, and the labels are orthogonal. Similar label representations hardly exhibit any interpretability during the training stage. Moreover, a phrase could also be regarded as a class label, where ambiguity is inevitable. As a result, two documents with the same label are likely to be from different domains and they only slightly overlap topics. In reality, the content of a document is often complex, and it is difficult to fully represent the document with only one label.

Therefore, we use multilabels as supervision signals to construct the semantic space in this work. On the one hand, it could cover more information than a single label thereby reducing ambiguity. On the other hand, the representation ability of the semantic space can be enhanced. However, there are few multilabel multilingual corpora that are directly available, and there are many multilanguage corpora that have the potential to become multilabel, such as Wikipedia.

To generate labels automatically, a quick way is to use Wikipedia concepts directly but the number of concepts is millions. It is possible to take advantage of linear methods to use all of them as classification labels. However it is not suitable to use millions of tags as an output layer of a deep neural network, and at the same time, the consideration of document connection in a multilabel manner is also difficult. Furthermore, because the classification labels are orthogonal to each other, it is also difficult to consider the natural connections between documents from the same language. Another route is to process the title and get the stem sequence as multilabels. This method is straightforward and efficient and has been used in many studies [8,29], but it also brings tens of thousands of labels. Moreover, the title is often concise and relatively general which would lead to uneven data distribution problems as a category label. Alternatively, adding multilabels manually is a feasible way. However, this approach is not only time-consuming and expensive but also difficult to generalize to corpora in other languages and domains.

Therefore, an automatic method must be employed to obtain multiple labels. The latent Dirichlet assignment (LDA) algorithm is a generative probabilistic model of a corpus and an unsupervised method to obtain the topic distributions of the document, which is widely used in natural language processing research. Thus, we could choose to use the LDA method to get supervision signals automatically, while assuming that the topic distributions obtained by LDA are sufficiently accurate. There are several advantages to doing so. First, multilabels can be automatically extracted from the data itself without additional information. Second, the inherent connections between documents in the same language could be involved. Third, it is easy to generalize to other languages. Fourth, the ambiguous interference of manual labels can be excluded. Fifth, the number of categories is controllable. Deep neural network methods can be exploited because of the flexibility of the number of categories. Moreover, according to the topic distributions returned by the LDA method, the number of categories and the number of multilabels could be adjusted. This also brings interpretability as each dimension corresponds to a topic with some vocabulary. For instance, assuming that the number of topics is given as 100 and the number of multilabels is set as 1, then the topic distribution by LDA is a 100-dimensional vector and their sum is 100%. The label is also a 150-dimensional vector, where the position of the topic with the highest probability is set to 1 and the remaining 99 are set to 0, which is a one-hot form. If the number of multilabels is set to 6, the label will then set the 6 topics with the highest probability to 1 and the remaining 94 topics to 0.

Learning from features to input data helps to improve the feature quality of classifier training, which will help to improve the discriminative ability of the shared semantic space. Lei’s work also proved that adding an unsupervised feature by auto-encoder could improve the performance of a linear classifier [30]. We follow this setting and use the supervised auto-encoder model. Denoting an encoder $f^{\left(l_{i}\right)}\left(x,\Theta^{i}\right)$ with it output $\hat{x}$, a decoder $g^{\left(l_{i}\right)}\left(\hat{x},Y^{i}\right)$, and an orthogonal matrix *H*, then the objective function as follows:(5)$$\begin{array}{cc} J^{\left(l_{i}\right)} & =\frac{1}{n_{i}}\sum_{j=1}^{n_{i}}\left[\lambda ∥H{\hat{x}}_{j}-y_{j}{∥}_{2}+\left(1-\lambda \right){∥g^{\left(li\right)}\left({\hat{x}}_{j},Y^{i}\right)-x_{j}∥}_{2}\right]\\ \; & =\frac{1}{n_{i}}\sum_{j=1}^{n_{i}}\left[\lambda ∥Hf^{\left(li\right)}\left(x_{j},\Theta^{i}\right)-y_{j}{∥}_{2}+\left(1-\lambda \right){∥g^{\left(li\right)}\left(f^{\left(li\right)}\left(x_{j},\Theta^{i}\right),Y^{i}\right)-x_{j}∥}_{2}\right] \end{array}$$
where λ is the trade-off parameter between reconstruction error and supervision loss. Each language is trained separately, and the gradient descent algorithm is used to iteratively search for the optimal parameters.

### 3.4. Implementation

The framework of MDL is summarized in Figure 2, including two main processes, automatic labeling, and MDL model training. First, concept ids and corresponding documents are extracted from a language-specific Wikipedia dump. The extracted document collection could be seen as a comparable corpus. Each document corresponds to only one concept id, but each concept id corresponds to multiple documents from multiple languages. Meanwhile, the comparable corpus is divided into a training set and a test set. The first process is labeling, a specified language is selected as the criteria and is used to construct the shared space. The topic distributions are automatically obtained through the LDA algorithm. The topic distributions of the training set are transformed into multilabels, which serve as supervision signals for the MDL model. The same supervisory signals are used by different languages, which are transferred by concept ids. The topic distributions of the test set are used to compute the cosine scores for document rankings, thus the retrieval results are obtained. The retrieval results are recorded by concept ids for transfer to other languages.

The second process is the training of the MDL model, and the Doc2Vec method is used for document representations (X) as the model input. X is transformed into the shared space ($\hat{X}$) by an encoder, then the features are used to predict labels (Y’) via the orthogonal matrix (H), which is a linear classifier. Document features ($\hat{X}$) are iteratively optimized in the shared space through the backpropagation of the supervisory signals. The decoder (g) could maintain the semantic consistency of the original language and improve the discrimination of features. Each language is trained individually to map documents to the semantic space. Thus, MDL reduces the amount of data during the training stage and also reduces time costs due to the parallel training. These findings lead to the conclusion that the proposed method reduces the time complexity and computational complexity.

## 4. Evaluation

This section demonstrates the performance of MDL embedding methods and compares them with current state-of-the-art models. The main evaluation is the accuracy of cross-lingual document retrieval, as MDL is designed for the representation of entire documents. We describe our experimental settings and show the main results, and analyze the effectiveness of our method calculation process.

### 4.1. Experimental Settings

**Dataset.** Wikipedia is used as the document collection because most of its articles exist in multiple languages, and each article is attributed to the language-independent concept it is about. For instance, both English “beer” and Italian “birra” are attributed to the concept Q44. Using the alignment between concepts, we could transfer labels among languages. In order to validate that the proposed method is language-independent, we have selected a few representative language pairs for the clarity of evaluation, including English (en), Italian (it), Danish (da), and Vietnamese (vi). English and Italian have more data and are high-resource language pairs, also, they have often been used in the prior literature [8,14]. Danish and Vietnamese were chosen due to their relatively small Wikipedias. Moreover, the cultural distance between Vietnamese and European languages is relatively far, and the intersection is small which increases the inclusiveness and robustness of MDL.

**Retrieval.** Our evaluation focus on cross-lingual document retrieval task, where we consider the entire documents as texts. For a pair of query and target languages, as well as query text, the objective is to return a sorted result of the target texts. The decreasing ranking is obtained by similarity computing in a shared embedding space. The main measure in this experiment is cosine, which is the most commonly used similarity measure. The mean average precision (mAP) is a common measure in IR [31] for calculating all the returned results of a comprehensive evaluation. MAP is defined as the average of retrieved precision of each query, also used as the evaluation metric in the experiments in the next sections.

**Baseline.** We consider the best-performing model of Josifoski et al. [2] (Cr5) as our main baseline. The Cr5 model has been shown to outperform other methods on the cross-lingual document retrieval task and is also a document-level cross-lingual representation method. We follow the settings of the CR5 model for preprocessing and then use the author’s code to retrain. We build vocabulary and count vocabulary frequency according to the same training data set of MDL. Words are discarded if their frequency is less than 3. In the testing stage, documents are represented according to model weights and term frequency weights.

**Data preprocessing.** Inspired by the work of Schwenk et al. [32], we downloaded Wikipedia’s search indices instead of Wikipedia dumps, https://dumps.wikimedia.org/other/cirrussearch/ (accessed on 27 December 2021), and extracted document ids, titles, document texts, and wikibase items, which contain raw text data and concept ids. The first hundred tokens of Wikipedia articles always summarize the full text [33]. Meanwhile, we limit document length from 50 to 1000, which covers most Wikipedia articles. This is very meaningful since it reduces the unnecessary computation caused by the text being super long, and also avoids the damage to the model due to the ambiguity and noise of the very short text. We tokenize text through the nltk toolkit [34], while the vocabulary is converted to lowercase letters and stop words are removed.

The document representation process is based on the data itself, so an unsupervised method is used to initialize the document embeddings. The widely used bag-of-words (BOW) model is simple and efficient, but the text structure information is not considered enough. Another way is based on the Tf-Idf algorithm, which represents documents by weighted word vectors and works well in many applications. However, this is a heuristic and not all document content can be included. We choose the doc2vec method to initial document representation [10]. The doc2vec method could contain text structure information and the document representations are optimized at the document level. It is assumed that this method can accurately reflect the textual features of a specific language. In order to avoid human ambiguity, we use an automatic way to generate retrieval answers. Multiclass labels can be generated by clustering methods, but it is a single label and contains insufficient supervision information. Thus, we used the LDA topic model to obtain the multitopic distribution of the documents, and then the top 30 are selected as reference answers based on the cosine distance ranking. In other words, the experiments simulate the monolingual retrieval process by using LDA and cosine methods. Thus, the number of correct answers can also be controlled, and the correct answers are still obtained automatically. The top 30 were chosen as the correct answers since the average number of relevant documents for most datasets is 30. Due to the limitation of the test languages and training data, we choose Wikipedia data as the experimental dataset and construct the data set for training and evaluation.

**Hyperparameters.** The proposed method would train multiple neural networks to handle the multilanguage data. The network is similar to a standard autoencoder, including an encoder and a decoder, and each module contains three fully connected layers with the rectified linear unit (ReLU) [35] activation function. The number of hidden units is 1024, and the number of output units of the encoder is 512. The orthogonal matrix H is randomly generated once which is used as a projective transformation. In the testing process, the decoder and the matrix H are ignored and the outputs of the encoder are the feature representations of the samples, which is from the shared common semantic space. The proposed model is trained on Nvidia GeForce RTX 3090 GPU with PyTorch. We use the ADAM [36] optimizer with a batch size of 256 and epochs are set to 50 for the training stage. The experimental results show that the performance of the model does not increase all the time as the amount of training increases. Therefore, to trade off performance versus computation time, the number of categories is set to 1000, the number of multilabels is set to 6 and λ is 0.5.

### 4.2. Document Retrieval

Our main evaluation is the accuracy of cross-lingual document retrieval. First, all texts are mapped to a predefined shared space, so as to obtain the semantic feature representations of multilingual documents. Second, the ranking result is obtained by calculating the correlation between document features. Finally, the mAP is calculated based on the ranking result. In this work, the evaluation is considered with two training settings, (1) joint training, and (2) pairwise training. Joint training uses the concept intersection of four languages as training data, and fits the models for any of the languages considered. At the same time, this could verify the transfer performance of the model among languages in multiple language scenarios. For example, achieving mutual retrieval between Italian and Vietnamese using English criteria. Pairwise training uses the concept intersection of two languages as training data to evaluate the retrieval performance of the proposed model. In the training stage, each language is trained individually and the output of the model is a language-specific encoder. The encoder transfers the language-specific initial document vector into a predefined shared space, afterward, the similarity between documents can be calculated no matter what language they come from.

#### 4.2.1. Joint Training

We training a multilingual model on all 4 languages, including English (en), Italian (it), Danish (da), and Vietnamese (vi), while testing on all 12 directed pairs. The dataset contains documents from all languages and is built based on Wikipedia data. First, the multilingual concept intersection is obtained based on the preprocessed Wikipedia data. Afterward, the concepts and corresponding documents that are too short and too long are removed. Keeping documents with lengths between 20 and 1000, finally, the number of concepts is 19,903. The statistics of the evaluation datasets are summarized in Table 1.

The data set is randomly shuffled, 1k concepts were selected as the test set and the rest are used as the training set. The experiment selects English as the criteria to generate multilabels through the LDA algorithm while the number of categories is set to 1000. MDL.1 indicates that the number of multilabels of documents in the MDL model is 1, which is equivalent to the single label. MDL.6 indicates the number of multilabels is 6. Table 2 summarizes the performance of our model in terms of mAP precision through the cosine similarity measure. It is observed that the performance of MDL.6 has at least a 30% improvement in mAP compared to the baseline method for bidirectional retrieval for each language pair. For high-resource language pairs such as English and Italian, the mAP of the Cr5 model exceeds 0.44, while MDL.6 reaches 0.57 under the same experimental settings, with a performance improvement of more than 29%. Furthermore, for low-resource language pairs such as Danish and Vietnamese, Cr5 achieves around 0.3 and MDL.6 achieves more than 0.55 on average where the improvement of performance is more than 80%. For the single-label models, the MDL.1 model outperforms the Cr5 model in this setting by over 20%. The reason is that a large number of redundant labels not only bring very limited positive effects but may even bring negative effects to the model. Thus, low-dimensional label sets (1 k) contain more useful semantic information as supervisory signals than high-dimensional label sets (1 m). The supervision signals play a very important role in the shared space, where MDL.1 model has gained a greater benefit in this experimental environment.

It is not enough to reflect the document representation ability of the model when the most relevant document can be retrieved, because the model may not understand the gaps between moderate relevant documents or between non-relevant documents. Some candidates will be misjudged because the retrieval conditions are too strict for the model. Thus, the overall position of all relevant documents in the ranking is compromised, and the so-called most relevant document may not be a precise answer. A better ranking result is placing all relevant documents first. The position of all relevant documents in the ranking can be used to evaluate the retrieval ability of the model. This could be shown by calculating mAP for different cutoff ranks, which is equivalent to adjusting the number *t* of retrieved documents. MAP is the mean of average precision (*AP*) where *AP* is calculated for one query as follows:(6)$$AP=\frac{\sum_{i=1}^{t}Relevant\left(i\right)*RelevantDocuments\left(i\right)/i}{N_{relevant}}$$
where Relevant(i)=1 if the document is relevant at rank *i*, and Relevant(i)=0 if not. RelevantDocuments(i) represent the number of relevant documents ranked less than or equal to *i*. $N_{relevant}$ represent the number of all relevant documents for the query.

The ranking ability of correct answers could be evaluated by different *t* settings. Figure 3 summarized the mAP scores where *t* is set as 10, 50, and 1000. The Cr5 model has a competitive accuracy at *t* = 10, which shows that the retrieval ability of the model is very strong since the most relevant documents can be ranked in the top 10 positions. Comparing all models, it is observed that the improvement between the MDL.6 model and the Cr5 model is more than 30% at *t* = 50 and 1000. The Cr5 model uses a single label as standard and anchor. It shows that the Cr5 model is too strict in sorting all relevant documents since the scope of supervision signal is not wide enough. The MDL.6 model enriches the document features due to the consideration of multilabel information. Thus, the ability to identify relevant documents is improved. It is observed that the accuracy of MDL.6 improves for any language pair when *t* = 50 or 1000. This shows that the MDL.6 model has learned richer semantic features than the baseline method so that all relevant documents are sorted as much as possible.

#### 4.2.2. Pairwise Training

For the scenarios where mutual retrieval of two languages is required, pairwise training was used in order to test the performance of the model on document retrieval tasks. With similar settings of joint training, pairwise training experiments using bilingual document intersection. Separate models are trained for all six language pairs of four languages, including en-it, en-da, en-vi, it-da, it-vi, and da-vi. The statistics of the evaluation datasets are summarized in Table 3. The number of intersections (265 k) between English and Italian is big as in high-resource languages, while the intersection size (32 k) between low-resource languages Danish and Vietnamese is relatively small.

The results of the pairwise training documents retrieval task are shown in Table 4. The number of categories is set to 200 and MDL.5 indicates that the number of multilabels is 5. $l_{1}$ is selected as the criteria for all the language pairs to build the shared semantic space. The influence of which language is selected as the criterion is not obvious in the retrieval results. Additionally, the $l_{2}$ criteria are discussed in the next section by the Danish and Vietnamese pair. Again, the performance has more than 30% improvement compared to the baseline method for each language pair. Even if it is an MDL model, multilabels are also better than a single label. Compared to joint training, the language retrieval performance for Danish and Vietnamese is worse. This is because joint training brings richer semantic information from high-resource languages to low-resource languages, which demonstrates the ability of knowledge transfer of the model. It also shows that joint training is beneficial to improve the retrieval performance of low-resource languages. The reason is that multilingual intersection will mask more noise, that is, leave more discriminative information and reduce ambiguous information.

To illustrate the performance, we also provide monolingual document retrieval results, taking the English (en) and Italian (it) pair as an example. Table 5 shows the performance comparison of cross-lingual retrieval and monolingual retrieval for MDL.5 and Cr5. In parentheses are the percentages of performance for cross-lingual retrieval versus monolingual retrieval. The cross-lingual performance of the MDL.5 model reaches 98% of monolingual retrieval, while the Cr5 model reaches 92%. The MDL model is closer to the results of monolingual retrieval. In addition, the monolingual retrieval performance of MDL.5 has a close 20% improvement compared to the Cr5 model. The reason is that Cr5 uses millions of labels as supervision signals but ignores the semantic relationships between labels. MDL models alleviate this problem by the use of multiple labels, which improves document representation across languages.

#### 4.2.3. Parameter Analysis

**Language Criterion.** As Figure 4 shows, there are 15 pairs of curves to express the performance trends of different language criteria, which are the Danish (da) and Vietnamese (vi) pair. Each pair of curves represents the mAPs for one model to evaluate multilabel numbers from 1 to 15, thus, the models are MDL.1, MDL.2,..., and MDL.15. Moreover, each curve is the average mAP of retrieval in both directions, including 20 categories settings, ranging from 50 to 1000. It is observed that each pair of curves is relatively close and the trends are similar. This indicates that the language used as the criteria to construct the shared common space has little influence on the retrieval performance of all the MDL models. Thus, $l_{1}$ was selected as the criterion by default for all experiments. This is because concept intersection has an equal status for both languages, the automatic labeling result of intersection documents is also similar, which of course also depends on the stability of the LDA topic algorithm and the avoidance of manual labeling ambiguity. Once the automatic labeling process is completed, the topic distribution of the document collection is settled, thus, the multilingual shared common space is settled. Even if the results of labeling are not necessarily the same every time, the shared space could map the documents with the same label closer, which provides the basis for the relevance calculation. Therefore, with different language criteria, the model performance is similar.

**Categories and number of multilabels.**Figure 5 shows the performance of pairwise training with the different number of categories and multilabels. Each line in the figure represents the trend of mAP value increasing with category where the number of multilabels is fixed. The number of multilabels is set from one to nine for all six language pairs, thus, each subplot has nine lines. It could be seen that the performance of the single label is the lowest, regardless of the language pair. However, the retrieval performance does not always increase with the number of multilabels. When the number exceeds 4, the model is pretty close to optimal performance. This shows that for Wikipedia articles with a length of 50 to 1000, using four multilabels is more effective than a single label, and especially it has advantages as supervision signals. Similarly, the retrieval performance does not always increase with the categories. For all language pairs, too few categories such as 50 are harmful to the model, while too many categories will not bring much growth and may degrade performance. For the bilingual retrieval task of Wikipedia data, the best results could be achieved without setting the number of categories greater than 1000. Based on this point, the MDL model reduces the supervision signal from millions of labels to thousands of labels, while maintaining the discriminative capacity of documents. The space is further compressed while retaining the representation ability of the shared space across languages.

**Supervised signals from LDA multilabels.** In order to verify the effectiveness of multilabel supervision signals, in other words, whether the document features have learned the information of the supervision signals, we generate multilabel representations through random numbers for the MDL model. Table 6 shows the results of random multilabels. The number of categories is set to 1000 and MDL.random5 indicates that the number of random multilabels is 5. The performance is similar for all language pairs in the experiments, and we choose English and Italian as a representative example. The performance of random labels is only about 0.1, which shows that it is not enough to construct a semantic space using supervised signals that only have differences but lack semantics. The topic distribution is automatically extracted by the LDA method not only contains semantic information, but plays an important role in the process of constructing the semantic space.

## 5. Conclusions

In this study, we propose a novel document representation approach (MDL) for cross-lingual documents retrieval task, which maps multilingual document features to the predefined shared semantic space. Cross language document representations are obtained through the individual learning of supervised autoencoders for each language. The strategy of automatic labeling for multilabel supervision signals increases the supervision information in the training stage while reducing artificial ambiguity in the semantic space. The MDL model enhances cross-lingual document features, thus, realizing the information transformation from high-resource languages to low-resource languages. Experiments on Wikipedia data show that the proposed method outperforms the state-of-the-art methods in cross-lingual document retrieval tasks with document-level representations. The MDL model still has two shortcomings that could be improved for retrieval tasks. In future work, the first aspect is to investigate how to enhance document features by integrating cross-lingual knowledge bases to improve retrieval performance. Another aspect is supervisory signals, since the multilabels used by the model are still a hard target, which contains the information on the correct labels. It is also very important for the document features to contain the information of wrong labels, which could reflect the difference in the document. Thus, how to integrate soft targets to construct the shared semantic space is a worthy problem to be solved.

## Figures and Tables

**Figure 1 entropy-24-00943-f001:**
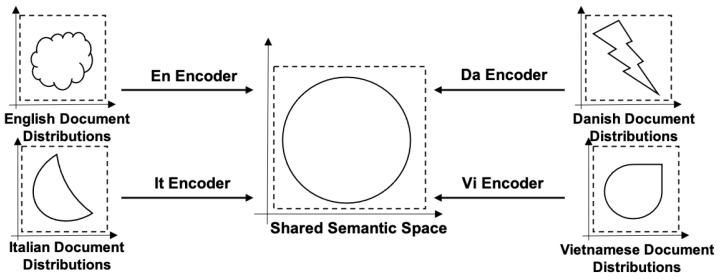
Multilingual document learning is defined as the mappings from multilingual documents to a shared semantic space. A language-specific encoder is the desired mapping, which contains the relation of multilingual documents to the language-independent semantic space.

**Figure 2 entropy-24-00943-f002:**
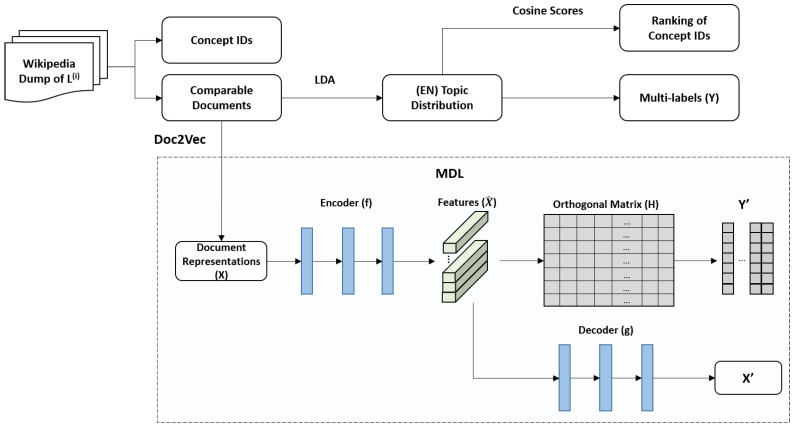
An overview of the learning process, including the model structure of MDL.

**Figure 3 entropy-24-00943-f003:**
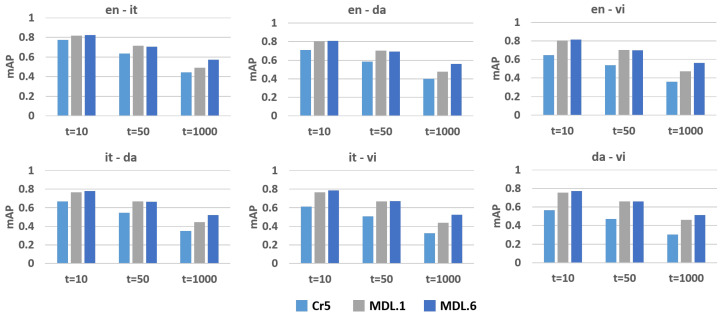
The mAP where the number of retrieved documents is set as 10, 50, 1000. Each pair was evaluated in both directions and the average is plotted.

**Figure 4 entropy-24-00943-f004:**
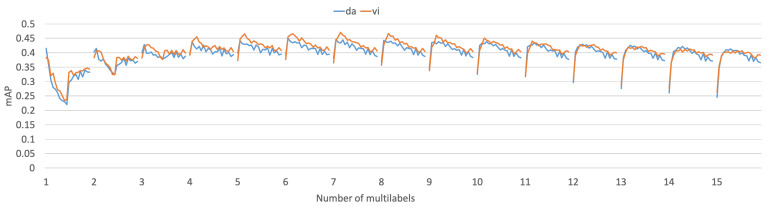
Performance comparison of document retrieval task for the two language criteria of the MDL model, which are Danish and Vietnamese.

**Figure 5 entropy-24-00943-f005:**
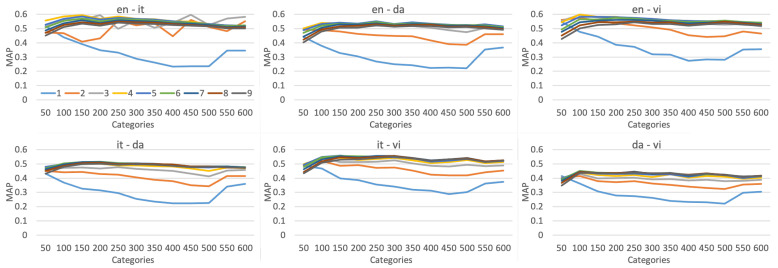
Performance of document retrieval task for the MDL model, with the different number of categories and multilabels.

**Table 1 entropy-24-00943-t001:** Basic statistics of Wikipedia data for evaluation, which is the concept intersection of four languages, including English (en), Italian (it), Danish (da), and Vietnamese (vi). Number of relevant represent average number of relevant documents per query.

Languages	en, it, da, vi
Document length	20–1000
Number of documents	19,903
Number of queries	1000
Number of relevant	30

**Table 2 entropy-24-00943-t002:** Performance comparison of joint training in terms of mAP scores. The number of categories is set to 1000 under which the correct answers are automatically constructed. MDL.1 is used as a single-label comparative experiment, and the number of multilabels is set to 1. The number of multilabels is set to 6 for MDL.6. The best results are highlighted by boldface.

	Cr5		MDL.1		MDL.6	
** $l_{1}$ $l_{2}$ **	**Query in $l_{1}$**	**Query in $l_{2}$**	**Query in $l_{1}$**	**Query in $l_{2}$**	**Query in $l_{1}$**	**Query in $l_{2}$**
en it	0.445	0.442	0.484	0.497	**0.57**	**0.578**
en da	0.401	0.4	0.464	0.49	**0.544**	**0.576**
en vi	0.37	0.347	0.464	0.48	**0.555**	**0.572**
it da	0.352	0.348	0.435	0.45	**0.514**	**0.531**
it vi	0.336	0.331	0.434	0.44	**0.521**	**0.528**
da vi	0.312	0.294	0.433	0.486	**0.519**	**0.652**

**Table 3 entropy-24-00943-t003:** Statistics of Wikipedia data for pairwise training. For 4 languages including English (en), Italian (it), Danish (da), and Vietnamese (vi), including 6 language pairs, en-it, en-da, en-vi, it-da, it-vi, and da-vi. Number of relevant represent average number of relevant documents per query.

Language Pair	en-it	en-da	en-vi	it-da	it-vi	da-vi
Document length	50–1000	50–1000	50–1000	50–1000	50–1000	50–1000
Number of documents	264,565	86,784	211,986	68,991	100,468	32,419
Number of queries	1000	1000	1000	1000	1000	1000
Number of relevant	30	30	30	30	30	30

**Table 4 entropy-24-00943-t004:** Cross-lingual documents retrieval performance of pairwise training in terms of mAP scores. The categories are set to 200. MDL.1 is a single-label comparison, and the number of multilabels is set as 1. The number of multilabels is set as 5 for MDL.5. The best results are highlighted by boldface.

	Cr5		MDL.1		MDL.5	
** $l_{1}$ $l_{2}$ **	**Query in $l_{1}$**	**Query in $l_{2}$**	**Query in $l_{1}$**	**Query in $l_{2}$**	**Query in $l_{1}$**	**Query in $l_{2}$**
en it	0.439	0.448	0.355	0.341	**0.573**	**0.576**
en da	0.399	0.4	0.379	0.381	**0.534**	**0.546**
en vi	0.37	0.348	0.442	0.445	**0.574**	**0.583**
it da	0.351	0.351	0.316	0.31	**0.507**	**0.515**
it vi	0.38	0.372	0.382	0.388	**0.54**	**0.555**
da vi	0.311	0.298	0.281	0.278	**0.419**	**0.439**

**Table 5 entropy-24-00943-t005:** The monolingual document retrieval performance of MDL.5 model and Cr5 model for English (en) and Italian (it). The table shows the mAP scores where the query language is l1 and the target language is l2. In parentheses are the percentages of performance for cross-lingual retrieval versus monolingual retrieval and the best results are highlighted by boldface.

$l_{1}$-$l_{2}$	en-en	it-en	it-it	en-it
Cr5	0.489	0.448 (91.6%)	0.476	0.439 (92.2%)
MDL.5	0.583	0.576 **(98.8%)**	0.581	0.573 **(98.6%)**

**Table 6 entropy-24-00943-t006:** Documents retrieval performance of random multilabels.

	MDL.random5		MDL.5	
$l_{1}$ $l_{2}$	**Query in $l_{1}$**	**Query in $l_{2}$**	**Query in $l_{1}$**	**Query in $l_{2}$**
en it	0.107	0.103	0.572	0.579

## Data Availability

The data is contained within the article.

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
