# Peer review of "Deep Multilabel Multilingual Document Learning for Cross-Lingual Document Retrieval"

_entropy, 2022, doi:10.3390/e24070943_

Round 1

Reviewer 1 Report

The paper addresses the problem of cross-lingual document retrieval, which is a topic of great interest. Although nicely structured, the presentation style could be improved. For example, the initial part of the Abstract is to verbose, and I would suggest the authors rewrite the Abstract and directly: (i) define the problem in a single sentence,(ii)  describe the proposed solution, (iii) discuss pros of proposed method over other methods, and (iv) summaries key results.

In the introduction, around line #51, some references indicating what are the competitive methods hinted by the authors should be provided. With respect to cross-lingual solution to exploit document retrieval, the authors may want to consider how researchers in the speech community have handled spoken document in different languages. For example, [ Siniscalchi, S.M. et al. "Exploiting Context-Dependency and Acoustic Resolution of Universal Speech Attribute Models in Spoken Language Recognition" Interspeech 2010 ] show how spoken documents in different languages can be mapped into the same vector space by leveraging universal speech attribute. In the Introduction, it is not clear what baseline is used to claim the 20% improvement. Other examples of poor presentation is when the authors give statements without providing supporting references for example: "The advantage of this method is that it is straightforward and has been used in many studies.." which studies ? Or when the authors inject sentence like "The concept intersection of the four languages is obtained through a python script first" in the middle of a section.

Some concepts discussed in the Related Work are repeated in the Introduction, e.g., the problem with translation-based methods. Such redundancy could be avoided. 

The proposed solution, in Section 3. Proposed Method and subsection, should be better described. The sentences that go from line #189 to #195 need to be revised. What do the author mean with "category labels are of equal status ?" I assume that Φ in Eq. (2) is specific for each language, and corresponds to the i-th encoder,  f^(l_i)(x,Θ^i); however, the fact the the encoder is not described yet makes a clear understanding of the method a bit complicated.  I believe the problem is that the authors rely too much on the fact that all readers are familiar with current neural representation methodology. I believe that the paper should be instead accessible to any reader, and that is the reason I suggest the presentation style be improved. Section 3.4. Implementation needs to be rewritten from scratch by properly describing each single block in Figure 2.

In the experimental section, the authors argue (see Table 2) that the proposed solution outperforms Cr5, but I wonder whether the improvement is statistically meaningful. By looking at Figure 3, I would say that Cr5 attains very competitive results at t=10 with a much simpler model. I wonder whether retrieving more than 30-40 documents in a cross-language setting is meaningful although it is clear the advantage of the authors' solution for those cases. 

Reviewer 2 Report

The authors present an original approach to cross-lingual document retrieval. The novelty is the way how the monolingual document representations are mapped into a shared space of concepts using LDA and deep learning. According to the authors, the results are SOTA and outperformed the referenced model Cr5. However, to dispel doubts, there are two major issues two address.

1. Results reported for Cr5 in Table 2 are much lower than those presented by Josifoski et al. (2019). The discrepancy is between 30-40 pp. Do you have any intuition why the performance of Cr5 in your research is much lower than the one reported in the original paper?

2. Did you use the pretrained Cr5 models for the baseline or trained them from scratch? If you trained them, please state it clearly in Section 4. I suggest adding a subsection "Baseline" for the description of how you retrained the models. However, if you use the pretrained models available at https://zenodo.org/record/2597441#.YqCaY3VBy-Y that might be problematic. If so, it is possible that the difference might be related to the data size/split.

Minor issues:

1. mAP reported in Tables 2 and 4 are for both languages? Could you provide what is the AP if you query in language X and the output for language Y? For instance, I would like to know what will be the AP for responses in English and Italian (separately) for a query in English. And the same for queries in Italian. 

2. In the introduction you mention an approach based on machine translation of the query. You stated, that this method is not flexible and depends on the MT quality. It sounds like this approach is insufficient. What improvements does your approach bring compared to the one based on query translation?

3. You stated that "(your) model is much efficient and easy to extended". Taking into account that the LDA defined the shared space of labels and it is built on a specified set of documents, for the new language you need a set of documents that are linked with the one used for building LDA. This is a kind of limitation. What is more, for a multilingual model (not only pair-wise) the number of documents might be limited. You mentioned that in the dataset you have 199k concepts. How many concepts are described in all four languages? Could you elaborate on what performance can be obtained by a model trained for all four languages at once?

4. How did you assure that there are exactly 30 relevant documents for each query? Is it 30 documents per language?

Round 2

Reviewer 1 Report

The authors have address all of my concerns.

Author Response

Thank you for the constructive and insightful advice to make the expression of this article more precise and logical.

Reviewer 2 Report

Thank you for the responses. They significantly improved the understanding of your work. However, there are still some statements, which require better justification or reconsideration.

Response 1.

If I understand it correctly, the mAP values reported in Table 2 are for t=1000? In the description there is "The category is set to 1000 (...)". Should it be changed to "The number of retrieved documents is set to 1000"?

Response 3.

Thank you for the results. Consider adding them to your articles as they are valuable to see what mAP can be achieved for cross-language retrieval. For example, for Italian, the mAP is almost the same when using the original query and query in English — 0.588 vs 0.573. 

Response 4.

Your response does not address what I meant. I just want you to support your statement: [40-42] "Therefore, large-scale translation in the Internet environment is impractical, also for some low-resource languages or domains which they do not contain enough data for training the machine translator, a more lightweight document representation is urgently needed." If you cannot support this statement by either citing other papers proving it or by your own evaluation, please remate this statement. 

Response 5.

Taking into account your response you should reformulate the statement: "Since each language is trained separately and takes document vectors as input, the model is much efficient and easy to extended." The statement "easy to extended" is overstated. Each model is trained separately, but you need the shared space of terms that are generated from parallel corpora. I would argue that training an MT model for a new pair of languages might be easier, as it does not require any shared space. Just a bilingual corpus. To handle many languages your approach requires a multilingual corpus. If this is not the case, please be more precise. 
